# User Recommendation for Data Sharing in Social Internet of Things

**DOI:** 10.3390/s21020462

**Published:** 2021-01-11

**Authors:** Kyoungsoo Bok, Yeondong Kim, Dojin Choi, Jaesoo Yoo

**Affiliations:** 1Department of SW Convergence Technology, Wonkwang University, Iksandae 460, Iksan, Jeonbuk 54538, Korea; ksbok@wku.ac.kr; 2Department of Information and Communication Engineering, Chungbuk National University, Chungdae-ro 1, Seowon-Gu, Cheongju, Chungbuk 28644, Korea; yeon5661@interpark.com (Y.K.); mycdj91@chungbuk.ac.kr (D.C.)

**Keywords:** social internet of things, data sharing, interaction analysis, similarity, user recommendation

## Abstract

As various types of data are generated on the social Internet of things (SIoT), which combine the Internet of things (IoT) and social networks, the relations of IoT devices should be established for necessary data exchange. In this paper, we propose a user recommendation scheme that facilitates data sharing through an analysis of an interaction between an IoT device and a user in the SIoT. An interrelation between a user and an IoT device as well as an interrelation between users exist simultaneously in the SIoT. Hence, the interaction between users must be analyzed to identify the interest keywords, and the interaction between IoT devices and users to determine the user’s preference of IoT device. Moreover, the proposed scheme calculates the similarity between users based on the IoT device preference based on IoT device usage frequency and interest keywords, which are identified through an analysis between the user and IoT device and that between users. Subsequently, it recommends top-N users who have a high similarity as the users for data sharing. Furthermore, the performance of the proposed scheme is verified through performance evaluation based on the precision, recall, and F-measure.

## 1. Introduction

The Internet of things (IoT) provides an intelligent environment that facilitates data exchange and communication between users and devices as well as between devices via communication modules attached on devices [1,2,3]. It connects things in the real world with the virtual world through networks to establish an environment in which communication between a human and an IoT device. Furthermore, it provides services such as surrounding environment monitoring, controlling, optimization, and autonomous operation through communication between a human and a device. As the amount of data generated and the number of things connected through communication on the IoT increase, the search and management costs for supplying data demanded by users increase as well. 

Social network services (SNS), which are used as online media for data sharing between users, facilitate the exchange of various data types through interactions between users or groups online [4,5]. SNS connects users through human networking and provides data to users through diverse community activities [6]. Recently, research has been conducted to provide necessary data to users through the integration of the IoT and SNS [7,8,9]. The social Internet of things (SIoT) has been developed to provide smart services to users through the integration of the IoT and SNS [10,11,12,13]. The IoT is generally used to generate and exchange data autonomously without the intervention of human beings. However, the SIoT is for combining both IoT environments and users [14,15,16]. On the SIoT, an interrelation between IoT devices is formed through social networks to provide necessary data to users. The Social Internet of Vehicles (SIoV) is an example of the SIoT applied to facilitate data sharing between vehicle drivers [17,18]. The SIoV provides various types of data based on the needs of drivers through communication between vehicles. Specifically, each device collects road and vehicle conditions and provides them to the driver.

In SIoT environment, the technologies are needed to easily obtain data required by the user. It is crucial to establish a social relation between IoT devices and users for selectively providing necessary data because users require different types of data in the SIoT environment [19,20,21,22]. Recommendation schemes have been proposed to expand social relation with IoT devices and users to exchange or share data. The existing recommendation schemes are classified into a scheme that recommends IoT devices that users can use and a scheme that recommends users to exchange necessary data. The device recommendation schemes have been proposed for data sharing between users and IoT devices by analyzing the relation of connection between devices used by users or the activities of similar users [23,24,25]. In [26], the device recommendation scheme considering time and space based on the relation between devices was proposed. However, this scheme fails to include distant devices because filtering is performed on devices based on a Receive Signal Strength Indicator (RSSI) in a wireless communication environment. In [27], a recommendation scheme based on the usage frequency of IoT devices was proposed, but this scheme did not consider the relation between users. In [28], a scheme for forming groups of similar IoT devices or people according to the data types requested by users was proposed to establish social relations for data sharing. However, this scheme calculates the similarity based on the data types requested by users, hence failing to consider the characteristics of both users and devices. In [29], the relation is efficiently re-establishes based on a clustering coefficient or a degree to reduce search cost. However, this scheme fails to consider the activities or interests of users. The existing schemes do not take into account interactions between various devices and users existing in the SloT environment, and do not reflect the characteristics of devices and users when calculating the similarity for recommendations.

In this paper, we propose a user recommendation scheme that facilitates data sharing based on social relations by analyzing the interaction between a user and a device, and that between users in the SIoT environment. The proposed scheme calculates the user’s IoT device usage by analyzing the interaction between users and IoT devices, and it evaluates that a user is deeply interested in a certain IoT device if the user uses it frequently. As users perform various activities through social networks, it analyzes interactions between users and identifies their interest keywords. It analyzes the similarity between users based on the preferences of IoT devices and interest keywords derived through interaction analysis and recommends users who possess a high similarity as users for data sharing. The contribution of the proposed scheme is as follows.

It recommends users who can share necessary data by considering both devices and users in SloT environments.It identifies a user’s IoT device preference which indicates the interaction degree between users and IoT device through the interaction analysis between users and IoT devices.It extracts a user’s interest keywords which appear frequently in social activities through the interaction analysis between users.It recommends new top-N users who have a high user similarity based on the IoT device preference and interest keywords.

This paper is organized as follows. In Section 2, the existing schemes for user or IoT device recommendation in the SIoT environment are reviewed. Accordingly, the problems pertaining to these schemes are described. In Section 3, the process of the proposed scheme is described. In Section 4, the excellent performance of the proposed scheme is verified through performance evaluation. In Section 5, conclusions and further research directions are presented.

## 2. Related Work

Previous studies proposed IoT device recommendation schemes that use the relation between a user and an IoT device to obtain useful data from IoT devices in the SIoT environment. The IoT device recommendation scheme was proposed to provide necessary data for users based on the distance between IoT devices and their relation [26]. This scheme compares the distance between IoT devices to the similarity between users to filter IoT devices. For IoT device filtering, an RSSI value between IoT devices and the similarity between them is calculated. If users who possess these IoT devices have similar interests, then these IoT devices are appropriate for data sharing. Hence, the similarity between IoT devices is calculated based on the frequency of data obtained from the IoT devices of users. After filtering the IoT devices, an IoT device is recommended based on the history of data exchanged between IoT devices using PageRank [30].

The relation between IoT devices should be analyzed for recommending an IoT device. The IoT device recommendation scheme proposed in [27] considers the time and space required for the use of IoT devices. If many people use two IoT devices frequently, it can be inferred that these IoT devices are similar to each other. In this regard, this scheme calculates the similarity between IoT devices using the similarity calculation scheme based on entropy proposed in [31]. The distance between IoT devices is calculated to consider the location of IoT devices and then used for IoT device recommendation. 

Similar user recommendation scheme for data sharing generate a group of users who have similar tendencies and allows them to obtain the desired data quickly in the group. The scheme presented in [28] forms a group of users or IoT devices similar to each other by generating user or IoT device profiles based on the data types requested by users. This scheme identifies a group related to the data requested by a user. If a group that can provide the requested data does not exist, it is then created through the calculation of the similarity based on node data. If such a group exists, then the user is included in the group.

The relation between users should be established to provide the data requested by users quickly in the SIoT environment. The scheme proposed in [29] recommends different users who can provide the data desired by users using information regarding their friends and the friends of their friends. In this scheme, it is assumed that users can be related to only certain friends. If the number of friends is below a threshold value, then the user is recommended instead of different users. This scheme suggests five strategies for user recommendation for data sharing. In the first strategy, a new user is not recommended when the number of friends related to a certain user is higher than a threshold value. In the second strategy, the degrees of friends are calculated to remove the relation with a friend who has the lowest degree. In the third strategy, opposite to the second strategy, the relation with a friend who has the highest degree is removed. In the fourth strategy, a clustering coefficient is calculated based on the number of nodes and edges of friends related to a certain user. Subsequently, the relation with a friend who has the lowest clustering coefficient is removed. In the fifth strategy, opposite to the fourth strategy, the relation with a friend who has the highest clustering coefficient is removed. 

In a SIoT environment, various schemes have been proposed to recommend IoT devices or users. In [26], IoT device recommendation through filtering was provided to reduce the calculation cost and time. In [27], a relation between IoT devices is established by considering the time and space used by users and calculates the similarity between IoT devices for IoT device recommendation. As a certain IoT device can provide different data types, the purposes for using this IoT device might vary. In [28], a group of users who are similar to each other is generated or expanded to provide data requested by a certain user. This scheme enables a user to gain access to data requested quickly in a group. When such a relation is disregarded, the precision of user recommendation for data sharing can decrease in the SIoT environment. In [29], a friend relation is generated by considering the relations of users. The scheme allows users to gain access to the desired data quickly by restricting the maximum number of connections among users and removing the most insignificant relation. In [26], IoT devices located far away are excluded because typical IoT devices are filtered based on RSSI values in wireless communication environments, such as those applying Wi-Fi or Bluetooth technologies. In [27], only the usage status of IoT devices without considering the properties of IoT devices are analyzed. In [28,29], the relation between a user and an IoT device during similarity calculation is not considered. In [29], the features of IoT devices or the interests of users are not considered and then a user who can provide significant data can be removed when numerous needs are requested. As numerous devices and users are connected to each other through networks in the SIoT environment, SIoT characteristics should be considered to recommend users who can share necessary data for the target user. However, the existing user recommendation schemes disregard the characteristics of IoT devices and SIoT and consider only the relations of users. Moreover, these schemes recommend users by generating simple profiles of users or IoT devices and calculating the similarity.

## 3. User Recommendation Scheme

### 3.1. Overall Procedure

In this paper, we propose a new user recommendation scheme to allow users with similar interests to share more data in the SIoT environment. The proposed scheme analyzes the interaction between a device and a user and that between users. It calculates the similarity by analyzing interactions between users as well as simple profile data. Specifically, it analyzes various social activities performed by users in social networks, such as document creation, review posting, and evaluation, to identify the interest fields of users. Moreover, users who use similar IoT devices are more likely to exhibit similar tendencies in terms of interests and hobbies. In this regard, the interrelation between the user and the IoT device used directly by the user should be considered. At this time, different types of IoT devices are regarded as the same IoT devices if the same data are used.

Figure 1 shows the overall structure of the proposed scheme. This scheme obtains data including the data types used by users, location of device providing data, and activity details of social network users to analyze the interrelation between users and IoT devices. In the User–Device (UD) interaction analysis, IoT devices used by users are analyzed to calculate the preference of IoT devices. If users use similar IoT devices, it can be inferred that they tend to be similar to each other. Hence, the frequency, date, and location of use of IoT devices are analyzed. In the User–User (UU) interaction analysis, the interests of users are extracted based on the activities of users performed in social networks. If documents created by users include similar keywords, it can be inferred that these users are more likely to possess similar interests. Hence, the similarity between users is calculated based on keywords extracted from documents created by users. The similarity, which are calculated through the analyses of interactions between users and IoT devices as well as the social activities of users, are combined to calculate the ultimate similarity and to recommend top-N users who possess a high similarity.

### 3.2. UD Interaction Analysis

In the SIoT environment, users can obtain and use different types of data based on various IoT devices. If users use the same data based on the same IoT devices, it can be inferred that they have similar characteristics. The device preference, which indicates the interaction degree between users and IoT devices, is derived through the UD interaction analysis. It is calculated based on the frequency and date of use of IoT devices. Users who have similar preferences of IoT devices are identified as those who possess similar tendencies and can be recommended as users for data sharing. If a user uses a certain IoT device frequently, it can be inferred that the user is deeply interested in the IoT device. In this regard, the usage frequency of IoT devices is proportional to the usage frequency of these devices by the users. The date of use of IoT devices is used to identify the recent interests of users. Based on the calculation of the difference between the current and latest times of use of these devices, if it is determined that the user has used IoT devices frequently, then it is inferred that the user is deeply interested in IoT devices. Subsequently, the preferences of IoT devices increases.

Figure 2 shows the UD interaction analysis process to identify a user’s device preference. First, the usage frequency of IoT devices is calculated based on the details of use of IoT devices by users. The importance degree of IoT devices for users can be calculated in a different manner based on the number of users possessing IoT devices and that of users using those devices. For example, it is assumed that an IoT device provides service using weather data obtained. If most users are interested in weather data, then they will have high usage frequencies of IoT devices providing weather data. In this regard, a comparison of the similarity may be meaningless. Meanwhile, if some users check weather data from an IoT device to perform their favorite activities every weekend, it can be inferred that they use the IoT device with more special interests and purposes. However, if the importance of IoT devices is determined using only the usage frequency of IoT devices, IoT devices that are used only once a week for special purposes cannot be identified as important IoT devices to users. Hence, the proposed scheme normalizes the usage frequency between the users and the devices and calculates the weight of the IoT device by considering the distribution of the device usage. When the weight of the IoT device is calculated, the user’s IoT device preference is calculated by comparing it with the importance of other users’ IoT devices.

The usage frequency of an IoT device *d* by the user *i* is calculated using Equation (1), where γ is a constant used to establish a time interval from the current point to a certain point for analysis and *n* is the usage frequency of the device during the γ time interval. $UT_{kd}$ refers to the difference between the current time and the time at which the device was used. $DU_{id}$ is calculated based on the usage frequency of IoT device. Equation (2) is the IoT device weight $DW_{id}$ which is used to compare the usage frequency of IoT devices of all users with the usage frequency of a certain IoT device *d* to calculate the device weight, where *k* is the number of all users. Equation (3) is used to calculate the IoT device preference, where *m* is the number of all IoT devices. $DP_{id}$ is the user’s device preference which is used to calculate the user similarity.
(1)$$DU_{id}=\sum_{k=1}^{n}\left(1-\frac{UT_{kd}}{\gamma }\right)$$
(2)$$DW_{id}=\frac{DU_{id}}{\sum_{j=1}^{k}DU_{jd}}$$
(3)$$DP_{id}=\frac{DW_{id}}{\sum_{j=1}^{m}DW_{jd}}$$

Figure 3 shows the normalization process of the usage frequency of IoT devices by users to determine the user’s device preference. For example, it is assumed that the usage of IoT devices *D*2 and *D*3 used by *U*1 based on the usage frequency of these devices are 2.1 and 82.3, respectively. When the usage of these devices are calculated based only on the usage frequency, the usage of the IoT device *c* will be higher than that of IoT device *D*2. However, when the IoT device weight is calculated and applied based on comparing the values of all users who use IoT devices *D*2 and *D*3 using Equations (2) and (3), the usage frequencies of IoT devices *D*2 and *D*3 are calculated to be 0.42 and 0.41, respectively. When the user weight is calculated and applied, the preference of IoT devices *D*2 and *D*3 are calibrated to be 0.51 and 049, respectively.

### 3.3. UU Interaction Analysis 

Users perform various social activities based on human relations through social networks established in the SIoT environment. A UU interaction analysis is performed to extract the interests of users who are not directly connected through social networks based on an analysis of their social activities. The proposed scheme is designed to recommend new users who can exchange data. In other words, a newly recommended user has neither a human relation nor a history of data exchanged with the target user. Interest keywords are extracted through analyses of documents created by users and their social activities, such as evaluations and document collection, to identify users for data sharing on the SIoT.

The interest keywords that appear frequently in documents created or collected by a user indicate the indirect interests of the user. However, if a keyword frequently used by the user is also used by all users, then it is unlikely to reflect the user interest. Hence, the scarcity of keywords should be considered to identify the interest keywords of users. To extract the interest keywords of a user, the significance of each keyword extracted from documents related to the user is evaluated. The significance of each keyword is calculated using Equation (4), where $KV_{ik}$ is the keyword weight of a keyword *k* extracted from documents related to a user *i*, and $KS_{ik}$ is the scarcity of the keyword *k*. If $D_{i}=\overset{n}{\underset{k=1}{\cup }}DK_{ik}$ is keywords extracted from all the documents created by a user *i*, the keyword weight is calculated using Equation (5). $DK_{ik}$ is defined as the frequency of a keyword. $SCR_{ik}$ is the frequency of a keyword *k* in documents created by the user *i*. As users tend to be interested in the documents they collect, a weight is applied to these documents. A logarithmic function is used to prevent the values of the documents from increasing significantly as a result of the weight applied.
(4)$$UI_{ik}=KV_{ik}\times KS_{ik}$$
(5)$$KV_{ik}=DK_{ik}{}^{1+\log SCR_{ik}}$$

The Term Frequency–Inverse Document Frequency (TF-IDF) [32] is used estimate the scarcity of a keyword *k* extracted from documents related to a user *i*. TF-IDF numerically represents the significance of a certain word in a certain document created by a user. The scarcity of the keyword *k* extracted from documents related to the user *i* is calculated using Equation (6). $DN_{i}$ refers to the number of documents created by the user *i*. If the usage frequency of the keyword is high, then the TF value increases. If the usage frequency of the keyword by other users is low, then the IDF value increases.
(6)$$KS_{ik}=\frac{DK_{ik}}{DN_{i}}+\log\frac{\sum_{j=1}^{n}DN_{j}}{\sum_{j=1}^{n}DK_{jk}}$$

Figure 4 shows the process for calculating the significance of a keyword extracted from documents related to a user. First, keywords are extracted from documents created and collected by users to calculate the usage frequency of each keyword. Subsequently, it is assumed that users *U*1, *U2*, and *U3* have used *K1* to *K8* keywords. Furthermore, it is assumed that each user has created four documents, and that each user has collected one to two documents. Table 1 shows the number of documents crated or scraped by users. *U*1 has created keywords *K1* and *K8* most frequently in the documents created by the user. However, the keyword *K7* is used more frequently than the keyword *K1* in the documents collected by the user. In this case, the number of keyword *K7* is higher than that of keyword *K1* in the documents collected despite the same usage frequency of these keywords. Consequently, the value of keyword *K7* is higher than that of keyword *K1*. Moreover, the scarcity of keywords is calculated using the TF-IDF scheme based on keywords used in the documents created by users. Keyword *K1* used in the documents created by *U*1 has a comparatively high use frequency, thereby resulting in a high TF value. However, the calculated IDF value of this keyword is low because this keyword has also been frequently used by other users. Consequently, the value of this keyword decreases because it does not significantly affect the identification of user interests compared with other keywords. Meanwhile, keyword *K7* in documents created by *U*1 is frequently observed in only documents related to the user. Hence, this keyword has a high scarcity value of 0.301. This scarcity value is used as the weight for the usage frequency of this keyword to derive the ultimate value of the keyword significance. 

### 3.4. User Recommendation

Since various IoT devices and users generate and exchange data in SIoT environments, the relation between users and IoT devices, and the relation between users coexist. It is necessary to determine the user similarity considering the interactions between users and devices, and those between users to recommend users. The proposed scheme extracts the IoT device preference and interest keywords that express the user’s tendency through UD interaction analysis and UU interaction analysis. To recommend Top-N users for data sharing, we calculate the user similarity based on the preference of IoT devices and the significance of the derived interest keywords. The user similarity $US_{ij}$ between *i* and *j* is calculated based on the device similarity $DS_{ij}$ and the interest similarity $IS_{ij}$ using Equation (7), where *α* is a weight value assigned to $DS_{ij}$ and $IS_{ij}$ and has a value from 0 to 1. The higher $US_{ij}$, the higher the similarity between between *i* and *j*. $DS_{ij}$ is to the similarity between devices used by users, which is calculated based on the preference of IoT devices derived through the UD interaction analysis, $IS_{ij}$ is the similarity between the interests of users, which is calculated based on the significance of interest keywords derived through the UU interaction analysis.
(7)$$US_{ij}=\alpha DS_{ij}+(1-\alpha )IS_{ij}$$

$DS_{ij}$ is calculated using Equation (8) as the similarity between IoT device preferences derived through the UD interaction analysis. $DS_{ij}$ is a modified cosine similarity adding the location of IoT devices to the general cosine similarity. The location of IoT devices that provide data should be considered when analyzing the relations between users and IoT devices. If users use IoT devices in the same location, then data sharing can be performed more smoothly. $CL_{ijk}$ is calculated using Equation (9) as the location similarity of IoT devices and while considering the location of IoT devices used and ranges from 1 to 2. $L_{ik}$ is the vector used to store the location data of an IoT device *d* used by a user *i*.
(8)$$DS_{ij}=\frac{\frac{\sum_{k=1}^{n}DP_{ik}\times DP_{jk}\times CL_{ijk}}{2}}{\sqrt{\sum_{k=1}^{n}DP_{ik}{}^{2}}\sqrt{\sum_{k=1}^{n}DP_{jk}{}^{2}}}$$
(9)$$CL_{ijk}=1+\frac{L_{ik}\cap L_{jk}}{L_{ik}\cup L_{jk}}$$

$IS_{ij}$ is the similarity between interest keywords derived through the UU interaction analysis. $IS_{ij}$ is calculated using Equation (10). As the social activities of users vary, the significance of keywords used by users who perform numerous social activities is high. On the contrary, the significance of keywords used by users who perform only a few social activities is low. $IS_{ij}$ is a Pearson correlation coefficient that calculates the similarity of the user interest regardless of their social activities by using the mean value $AVG_{i}$ of the interest keywords of users.
(10)$$IS_{ij}=\frac{\frac{\sum_{k=1}^{m}\left(\left(UI_{ik}-AVG_{i}\right)\times \left(UI_{jk}-AVG_{j}\right)\right)}{2}}{\sqrt{\sum_{k=1}^{m}{\left(UI_{ik}-AVG_{i}\right)}^{2}}\sqrt{\sum_{k=1}^{m}{\left(UI_{jk}-AVG_{j}\right)}^{2}}}$$

In a SIoT environment, users and recommended users should have share data and mutual access to necessary data. To share data among users, we first need to establish a human network with the recommended users. Even if a user is connected by a human network, certain users may not provide their data to other users. Therefore, the user and the recommended users connected by the human networks should be allowed to share their data. In addition, if there is a large amount of data provided by users connected to the human network, it takes a lot of time to find the desired data, so it should be filtered and provided based on the user’s interests. Algorithm 1 shows a data sharing API with recommended users in a SIoT environment. For data sharing with the recommended users, we first perform a *relationship*( ) to perform a human network setup for each user *i* included in the *recUC*. We stop sharing data if there is no human network with the recommended users. If the human network is constructed through the *hrelationship*( ), we perform a data sharing request to the user through *reqDatasharing*( ). When the data sharing request is accepted, *filterData*( ) is performed to preferentially collect and share data similar to the user’s interests with the recommended users.
**Algorithm 1***DataSharingAPI(recUC, target)*// *recUC is recommended Top-N users*// *target is a user receiving recommended users*
*for each i included in recUC do* *i*
*← readUC(recUC)* *if hrelationship(i, target) then*  *shDataset*
*← reqDatasharing(i, targetU)*  *fDataset*
*← filterData(shDataset, targetU)*  *receive(fDataset, target)* *end if**end for*

## 4. Performance Evaluation

To verify the performance of the proposed scheme, it was compared with existing similarity calculation schemes by recommending the most similar users to the target user based on practical data and randomly generated data. An experiment was conducted using the Eclipse program based on the Java language under the experimental conditions of a Windows 7 64-bit operating system, Intel core i5-6400 CPU 2.70 GHz with 16 GB of memory. As a practical dataset related to the SIoT was not provided, the IoT network dataset provided in [33] was used for the experimental evaluation. The types, brands, and model names of devices that can be used by 4000 users and the service types that can be used based on the device type by analyzing devices possessed by 50,000 users are provided in [33]. The proposed scheme extracts keywords to identify user interests. Users can perform various social activities on the social IoT, but they cannot derive their interests from all social activities. In general, users scrap documents created by others to create their own documents to present their opinions or to share the documents they are interested in. No dataset exists to provide users with social activities in a SIoT environment. Therefore, user-generated documents and scraps are arbitrarily assigned to the dataset provided by [33] so that each user can determine their interest. We randomly generated social activities in which each user created 1 to 30 documents and scraped 0 to 10 documents over 30 days. Table 2 shows a dataset used for performance evaluation.

In the proposed scheme, the weight α is applied to the similarity between devices and that between user interests to calculate that between users using Equation (7). The weight α affects the user similarity as well as the precision of user recommendation. Hence, in this paper, the performance was evaluated by adjusting the α value from 0.2 to 0.8 and increasing the number of top-N users from top-5 to top-30 at intervals of five to identify the optimal α value used in Equation (7). Accordingly, the precision based on the weight value was analyzed. Figure 5 shows the result of analyzing the precision based on the weight value. Through performance evaluation, it was confirmed that the weight of 0.7 resulted in the highest precision. This result indicates that a more precise result is derived when a higher weight is applied to the device preference than to that of user interest. Hence, the proposed scheme was compared with existing schemes based on α value of 0.7, which resulted in the highest precision.

The performance of the proposed scheme was analyzed through a comparison with the performances of existing similarity calculation schemes. Hence, precision, recall, and F-measure were used. Precision refers to the ratio of users included in the correct answer set among the recommended users. Recall refers to the ratio of users who were included in the correct answer set and recommended as similar users. F-measure refers to the harmonic mean of precision and recall. There are IoT device recommendation schemes and user recommendation schemes as ones for sharing data in the SIoT environment. IoT device recommendation schemes take into account the history, location, and usage time of an IoT device and do not consider users’ social activities. User recommendation schemes create groups between similar users or devices depending on the query type, or calculate similarities by considering the connectivity of users. However, they do not consider a connection between the user and the device or the user’s social activities. In particular for [29], which is most similar to the proposed scheme, the process of determining user similarity was not presented. We conduct performance evaluation by adding frequency of use of IoT devices and frequency of keywords extracted from documents used by the existing recommendation schemes to perform their performance evaluations in environments similar to the proposed scheme. 

In order to compare performance differences according to the method of determining the similarity between users, we compared the top-N recommended using similarity measures such as Cosine similarity, Pearson similarity, and Minkowski distance. At this time, the existing similarity calculation schemes disregarded the characteristics of IoT devices and considered only the usage frequency of IoT devices by users as well as that of keywords in documents. The cosine similarity generates a value of −1, 0, or 1 based on the usage frequency of IoT devices by users, as shown in Equation (11). In this scheme, −1 is derived when the directions of vectors are opposite to each other, 0 when they are independent of each other, and 1 when they are the same. In Equation (11), $F_{ik}$ and $F_{jk}$ refer to the usage frequency of the IoT device *k* by users *i* and *j*, respectively. The Pearson similarity generates a value of 1 and -1 when the similarity is the highest and lowest, respectively, using a correlation coefficient between two vectors. Equation (12) is the Pearson similarity, where $AVG_{i}$ and $AVG_{j}$ is the mean usage frequency of IoT devices by users *i* and *j*, respectively. The Minkowski distance is used to generalize the Manhattan and Euclidean distances. As shown in Equation (13), the Minkowski distance calculates the difference between the usage frequency of device d by users *i* and *j*.
(11)$$CS_{ij}=\frac{\frac{\sum_{k=1}^{n}F_{ik}\times F_{jk}}{2}}{\sqrt{\sum_{k=1}^{n}F_{ik}{}^{2}}\sqrt{\sum_{k=1}^{n}F_{jk}{}^{2}}}$$
(12)$$PS_{ij}=\frac{\frac{\sum_{k=1}^{n}\left(\left(F_{ik}-AVG_{i}\right)\times \left(F_{jk}-AVG_{j}\right)\right)}{2}}{\sqrt{\sum_{k=1}^{n}{\left(F_{ik}-AVG_{i}\right)}^{2}}\sqrt{\sum_{k=1}^{n}{\left(F_{jk}-AVG_{j}\right)}^{2}}}$$
(13)$$MD_{ij}={\left(\sum_{k=1}^{n}{\left|F_{ik}-F_{jk}\right|}^{p}\right)}^{1/p}$$

Figure 6 shows the results of precision calculated using the proposed scheme and existing similarity calculation schemes based on simple processes. The number of users to be recommended was increased gradually at intervals of five during the precision calculation. The mean precision was calculated to be approximately 39% to 44% in the existing schemes, whereas it was calculated to be approximately 70% in the proposed scheme. This result indicated that the proposed scheme resulted in a higher precision than the existing schemes, and that the performance of the proposed scheme increased by 158% to 177% compared with those of existing schemes in terms of precision. This result was obtained because the existing schemes calculated the similarity with focus on frequently used IoT devices. As some IoT devices such as devices for providing weather data were used daily, these schemes considering only the usage frequency of devices failed to correctly calculate the similarity. Meanwhile, the proposed scheme considered the characteristics of these IoT devices and hence resulted in a high precision.

Figure 7 shows the results of recall calculated using the proposed scheme and existing similarity calculation schemes based on simple processes. The number of users should be gradually increased at intervals of five during the experiment. The mean recall was calculated to be approximately 35% to 45% in existing schemes, whereas it was calculated to be approximate 71% in the proposed scheme. This result indicated that the performance of the proposed scheme increased by 156% to 199% compared with those of existing schemes in terms of recall. This result was obtained because the proposed scheme identified similar users who were included in the correct answer set as the number of users to be recommended increased. Meanwhile, the existing schemes failed to identify similar users who were included in the correct answer set as the number of users to be recommended increased. Consequently, the excellent performance of the proposed scheme was validated in terms of recall based on the comparatively higher recall derived. 

Figure 8 shows the F-measure calculated using the proposed scheme and existing similarity calculation schemes based on simple processes. The number of users should be increased gradually increased at intervals of five during the experiment. The F-measure was calculated to be approximately 31% to 41% in the existing schemes, whereas it was calculated to be approximately 65% in the proposed scheme. This result indicated that the performance of the proposed scheme increased by 157% to 191% compared with those of existing schemes in terms of the mean F-measure. As the existing schemes considered only the usage frequency of devices, they were unlikely to identify similar users. On the contrary, the proposed scheme considering the characteristics of IoT devices resulted in an increase in precision, recall, and F-measure.

## 5. Conclusions

In this paper, we proposed a new user recommendation scheme similar to the target user through interaction analysis between IoT devices and users in SIoT environments. The proposed scheme was proposed to analyze interactions between users and IoT devices to facilitate efficient data exchange between users. During the UD interaction analysis, the proposed scheme calculated the preferences of IoT devices used by users by considering the characteristics of IoT devices and estimated the similarity based on the action value obtained. Moreover, the social activities of users were analyzed to identify the fields of their interests based on the details of their activities through SNSs. As the users were more likely to create or collect documents related to their interests, keywords were extracted from documents created or collected by users. The keyword value was calculated based on the usage frequency of the keyword extracted and its scarcity, and the similarity was estimated based on the keyword value obtained. Finally, the similarity derived through the aforementioned analyses were combined to recommend top-N users who had the highest similarity. Furthermore, the proposed scheme was compared with existing similarity calculation schemes through performance evaluation. The proposed scheme yielded an increase in precision by approximate 158% to 177%, an increase in recall by approximately 156% to 199%, and an increase in F-measure by approximately 157% to 191% compared with the existing schemes. The precision of user recommendation results derived from the proposed scheme increased because the characteristics of IoT devices were considered. The proposed scheme improves performance compared to the existing schemes, but it suffers from performance degradation if the number of recommended users is small. In the process of interaction analysis, it is also only considering the characteristics of extracting keywords of interest from users’ social activities and have failed to consider human network types. In the future, we will conduct research on improving performance even if the number of recommended users is small and on recommendation schemes considering more diverse social activities. 

## Figures and Tables

**Figure 1 sensors-21-00462-f001:**
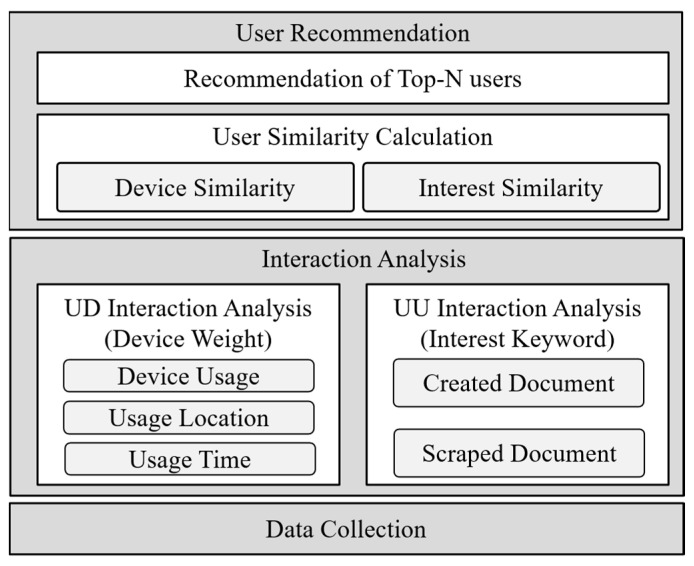
Overall structure of the proposed scheme.

**Figure 2 sensors-21-00462-f002:**
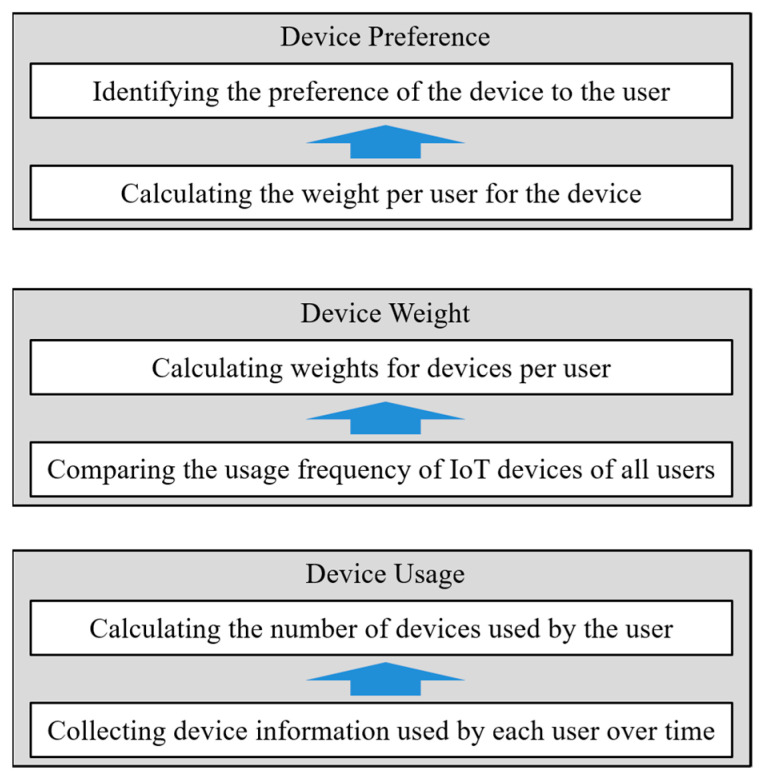
UD interaction analysis process.

**Figure 3 sensors-21-00462-f003:**
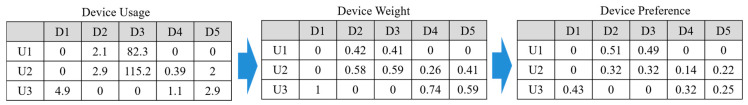
Examples of normalization of action values.

**Figure 4 sensors-21-00462-f004:**
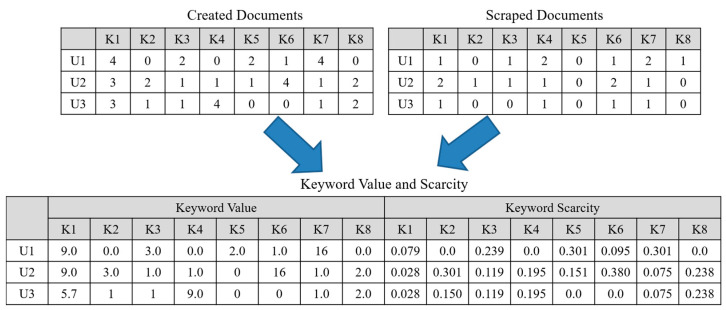
Examples of keyword values extracted.

**Figure 5 sensors-21-00462-f005:**
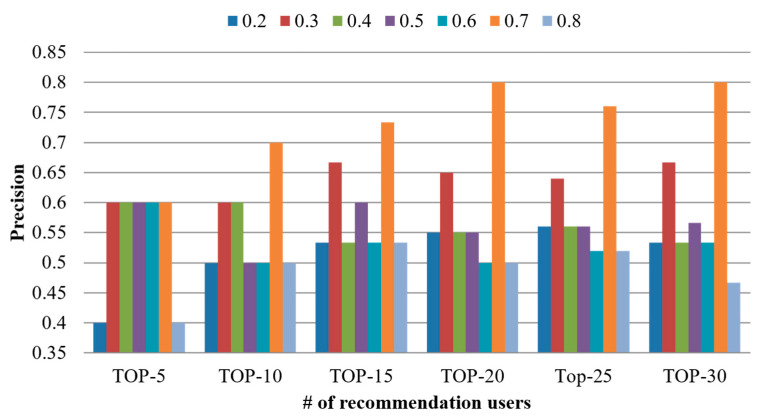
Comparison of precision based on weight.

**Figure 6 sensors-21-00462-f006:**
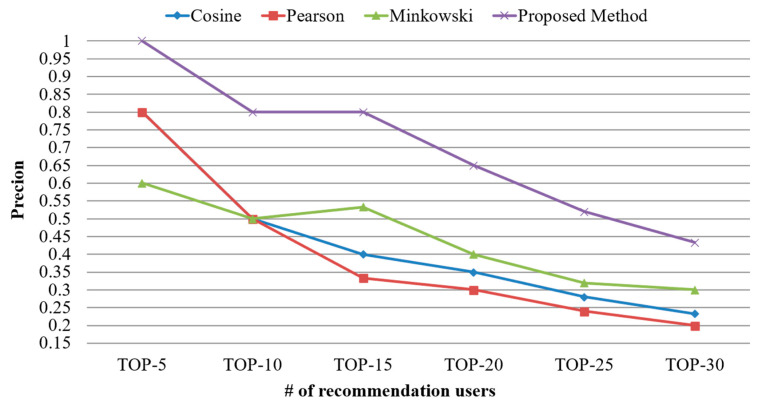
Comparison of precision based on the number of top-N users.

**Figure 7 sensors-21-00462-f007:**
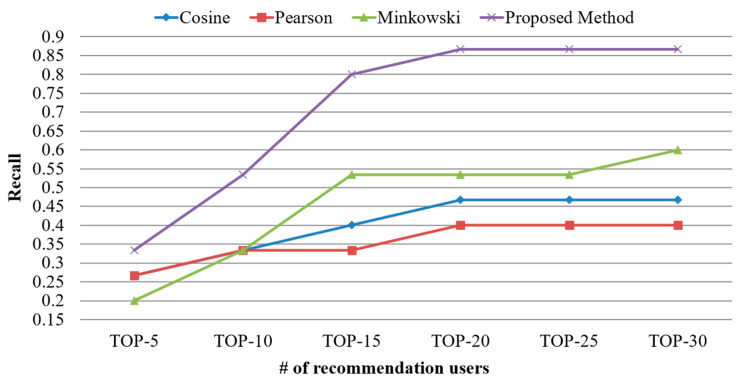
Comparison of recall based on the number of top-N users.

**Figure 8 sensors-21-00462-f008:**
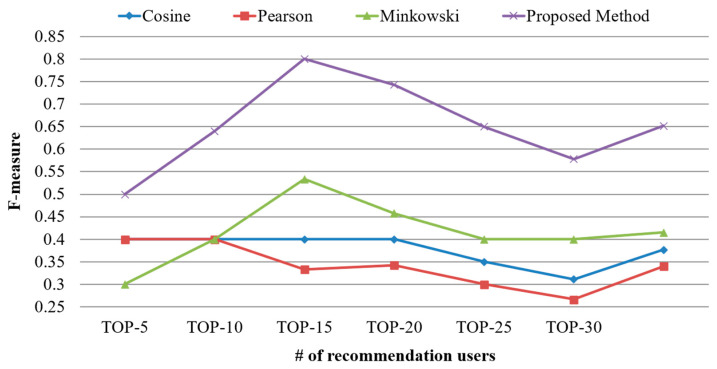
Comparison of F-measure based on the number of top-N users.

**Table 1 sensors-21-00462-t001:** The number of Documents.

User	# of Created Documents	# of Scraped Documents
U1	4	2
U1	4	2
U3	4	2

**Table 2 sensors-21-00462-t002:** Dataset for performance evaluation.

Parameters	Values
Number of users	4000
Number of IoT devices	18
Measurement period	30
Number of user created documents	0~30
Number of scraped documents	0~10
Weight (α)	0~1
Number of recommendations	5~30

## Data Availability

Publicly available datasets were analyzed in this study. This data can be found here: http://www.social-iot.org/index.php?p=downloads.

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
