# Peer review of "User Recommendation for Data Sharing in Social Internet of Things"

_sensors, 2021, doi:10.3390/s21020462_

Round 1
Reviewer 1 Report
Paper deals with important task. It has a logical structure, all necessary sections.
Suggestions:
-
- Main contribution is weak. Authors should present it in point-by-point scheme (2-3 items) and clearly indicates the novelty of this paper.
- Authors should argue their choice on similarity distances used for modeling.
- It would be good to see the comparison with other similar schemes.
- Authors should add a limitations of the proposed approach in the Conclusion section.
Author Response
Dear Reviewer,
We would like to sincerely thank you for your attentive indications and good comments. Our paper is partially rewritten in order to revise and complement your comments. Please refer to the attached file about the detailed revisions.
Many thanks.
Jaesoo Yoo

Reviewer 2 Report
The paper describe a user recommendation scheme that facilitates data sharing through an analysis of an interaction between an IoT device and a user on the SIoT
Please review the sintax for the next phrases:
(line 194)
However, the usage frequency of the IoT device will be low owing to the low usage frequency, as it is used only once per week.
Line (197)
The weight of an IoT device is used to identify the distribution of use of the device based on the usage weight of the device. The device preference is used to identify the importance of a certain IoT device based on the weight of the IoT device by the use.
The authors could you to describe with more detail the social data, because they explaine the general idea related to use a random function for generate the data of 1~30 documents.
For the data exchange between users would be interesting to include in the model the related to permiss for sharing documents and suggest API for access to information from user social networks
Author Response

(The authors gave the same response as above.)

Reviewer 3 Report
This paper aims to propose a scheme to analyse the interaction between users to identify the interest keywords, and the interaction between IoT devices and users to determine the user’s preference of IoT.
The introduction section has provided a decent overview on the challenges supposed to be tackled be the proposed scheme, although it could be introduced in a less complicated way.
In section 2 the related works have been reviewed, however, it misses an appropriate conclusion and point of departure(POD). There are some POD in section 3 which could be move to the end of section 2 as the POD for the reviewed related work.
Author Response

(The authors gave the same response as above.)
